# Selective Recovery of Copper from a Synthetic Metalliferous Waste Stream Using the Thiourea-Functionalized Ion Exchange Resin Puromet MTS9140

**Alex L. Riley** [1,2,*], **Christopher P. Porter** [1,3] and **Mark D. Ogden** [1]

1    Separations and Nuclear Chemical Engineering Research (SNUCER) Group, Department of Chemical and Biological Engineering, University of Sheffield, Sheffield S1 3JD, UK; chris.porter@wastecare.co.uk (C.P.P.); m.d.ogden@sheffield.ac.uk (M.D.O.)
2    Department of Geography, Geology & Environment, University of Hull, Hull HU6 7RX, UK
3    WasteCare Ltd., Richmond House, Leeds LS25 1NB, UK
*    Correspondence: a.l.riley@hull.ac.uk

**Abstract:** The extraction of Cu from mixed-metal acidic solutions by the thiourea-functionalized resin Puromet MTS9140 was studied. Despite being originally manufactured for precious metal recovery, a high selectivity towards Cu was observed over other first-row transition metals (>90% removal), highlighting a potential for this resin in base metal recovery circuits. Resin behaviour was characterised in batch-mode under a range of pH and sulphate concentrations and as a function of flow rate in a fixed-bed setup. In each instance, a high selectivity and capacity (max. 32.04 mg/g) towards Cu was observed and was unaffected by changes in solution chemistry. The mechanism of extraction was determined by XPS to be through reduction of Cu(II) to Cu(I) rather than chelation. Elution of Cu was achieved by the use of 0.5 M–1 M $NaClO_3$. Despite effective Cu elution (82%), degradation of resin functionality was observed, and further detailed through the application of IC analysis to identify degradation by-products. This work is the first detailed study of a thiourea-functionalized resin being used to selectively target Cu from a complex multi-metal solution.

**Keywords:** copper; ion exchange; thiourea; X-ray photoelectron spectroscopy (XPS); reductive extraction; resource recovery





## 1. Introduction

Copper is a globally indispensable metal used in a wide range of industries, especially in electronics, and the demand for copper resources have driven production rates exponentially over the past century [1]. Copper is traditionally obtained through the extraction of natural ore deposits, predominantly chalcopyrite ($CuFeS_2$) and chalcocite ($Cu_2S$), and subsequent processing by pyro- and hydro-metallurgical processes. However, these processes are not entirely efficient and result in unrecovered copper remaining present in flotation tailings [2]; a source which is predicted to increase as high-quality ore deposits are progressively depleted [3]. Furthermore, copper ores also contain metals such as Co, Ni, and Zn, and are associated with minerals such as pyrite ($FeS_2$) and sphalerite (ZnS) [4]. As such, it is possible for copper process waters to contain an abundance of other metal species, highlighting the need for the selective recovery of copper away from other first-row transition metals.

In addition to mineralogical sources of copper, recent efforts have been made in recovering copper from novel sources, including industrial residues [5], legacy waste deposits [6], sewage sludge [7], and e-waste [8]. Of these sources, e-wastes (particularly printed circuit boards (PCBs)) have received significant attention for their resource recovery potential [9–12]. Specifically, spent acid etching solution and waste PCB sludge have been candidates for resource recovery [9], with the sludge containing Fe, Cu, Ca, Sn, Al, Zn, and Cr [10]. Should a hydrometallurgical approach be taken to liberate these metals from

PCB wastes, it follows that a complex waste stream containing multiple metals would be produced. As such, the need for an effective method of selectively recovering copper away from other metals present in solutions is further emphasised.

As a complexing reagent, thiourea ($SC(NH_2)_2$) is most commonly used during the hydrometallurgical leaching of gold from ores as a less-toxic alternative to cyanide [13,14]. Further uses include the extraction of gold and silver from PCBs [15], and, more recently, it has been explored as part of a dual-lixiviant treatment process for waste activated carbon [16]. Given its high affinity for metals as a ligand, thiourea has been commercially incorporated into solid-phase extraction media, particularly ion exchange resins, though to date it has only been applied for precious metals extraction [17]. This paper describes the results of a previously unexpected interaction exhibited by the thiourea-functionalized resin Puromet MTS9140. Initially performed as part of a mixed-metal screening experiment to assess resin metal extraction behaviour, a high selectivity towards copper was observed and further investigated, the results of which are presented here in detail for the first time.

## 2. Materials and Methods

### 2.1. Solution Preparation

A mixed metal stock solution was prepared using the sulphate salts of Al(III), Co(II), Cu(II), Fe(III), Mn(II), Ni(II), and Zn(II). All metal salts used were of analytical grade and purchased from Sigma-Aldrich. Salts were dissolved in deionized water and acidified to pH 1 using $H_2SO_4$ such that the final concentration of each metal was 2000 mg/L. While concentrations in real waste leachates would be heterogeneous, the use of equal concentrations in this work ensured that observed differences in metal extraction were the result of resin behaviour. The resulting products of this "stock solution" were taken and diluted to produce the pregnant liquor solutions (PLS) that were used in experimental procedures (typically 200 mg/L). The PLS were adjusted to the appropriate pH using concentrated $H_2SO_4$ and NaOH to minimize changes in volume. Working from a stock solution ensured continuity in metal concentrations between batches of PLS, hence minimising variation in solutions between resin contacts. For the pH screening study, addition of $H_2SO_4$ allowed a range of acidities to be explored, from 0.01–3 M $H^+$. Where the effects of increased sulphate concentration were being tested, this was achieved by addition of $(NH_4)_2SO_4$, ranging from 0.02–4 M $SO_4^{2-}$. Eluent solutions were prepared by dissolution of $NaClO_3$ and adjustment to pH 2 using 37% HCl.

### 2.2. Resin Preconditioning

Puromet MTS9140 was supplied by Purolite International Ltd. and was preconditioned through contact with excess 1 M $H_2SO_4$ (S:L 1:50) for 24 h while being agitated on an orbital shaker. Five washing cycles using excess deionized water (S:L 1:50) ensured the removal of residual acid from the preconditioning process, and the resin was stored under deionized water until required. Samples of resin were dried at 50 °C to determine its density, which is provided in Table 1 alongside the manufacturer specifications.

**Table 1.** Manufacturer specifications of Puromet MTS9140 from Purolite International Ltd. (PS-DVB = Polystyrene cross-linked with Divinylbenzene).

| Functional Group | Capacity (eq/L) | Polymer Matrix | Moisture (%) | Particle Size (µm) | Density (g/mL) |
|---|---|---|---|---|---|
| Thiourea | 1 | PS-DVB [1] | 50–56 | 300–1200 | 0.308 |

[1] PS-DVB = Polystyrene crosslinked with Divinylbenzene.

### 2.3. Static Equilibrium Experiments

Static (batch) experiments were performed by contacting a fixed volume of resin with a constant volume of solution and allowing the system to equilibrate. Solutions were generated such that the effect of a range of acidities and sulphate concentrations on metal extraction could be determined.

Resin was measured out volumetrically by pipetting the resin/water slurry into a measuring cylinder, inverting the cylinder to promote particle size mixing, and allowing the resin to settle under gravity (herein referred to as 'wet settled resin'). 2 mL of wet settled resin was drained and contacted with 50 mL of PLS. Containers were placed on an orbital shaker and contacted for 24 h to equilibrate, after which the supernatant pH was measured using a calibrated silver/silver reference electrode and sampled for elemental analysis. This was achieved through dilution using a 1% $HNO_3$ solution prior to analysis via inductively coupled plasma optical emission spectroscopy (ICP-OES; Perkin Elmer Optima 5300 DV or Spectro Arcos model) or flame atomic absorption spectroscopy (AAS; Perkin Elmer AAnalyst 400 model). For all instruments, regular check-standards were analysed to ensure data accuracy, and instruments were recalibrated if readings were beyond 2.5% of the expected standard concentrations.

## 2.4. Fixed-Bed (Dynamic) Experiments

For dynamic breakthrough experiments, small-scale columns were completely packed with resin and capped at both ends with Teflon frits, resulting in a total bed volume (BV) of 5 mL wet settled resin. The columns were agitated during packing to promote homogeneous distribution of resin particle size throughout the bed. To ensure efficient mass transfer between solution and resin and to reduce the risk of 'channelling' [18] a reverse-flow setup was employed, whereby the PLS was introduced at the bottom of the vertical column. For elution studies, a smaller BV of 1.4 mL of wet settled resin was used to minimize the concentration of eluent peaks and required dilution for analysis. PLS were pumped using either a 'Heidolph Hei-Flow Value 01' pump with 'SP Quick' pump head, or a 'BioRad Econo Gradient Pump'. Verification of solution flow rates was achieved by pumping deionised water through each packed column for a set time and using the mass of water collected to calculate volumetric flow in bed volumes per hour (BV/h). Effluent solutions were collected using a 'BioRad Model 2110' fraction collector set to advance at specified time intervals and diluted using either 1% $HNO_3$ for ICP-OES or AAS analysis, or deionized water for ion chromatography (IC) analysis. IC analysis was performed using a Metrohm '833 Basic IC Plus' fitted with a 'Metrosep A Supp 5' column (PVA-quaternary ammonium), a carbonate eluent (4.5 mM $Na_2CO_3$, 803 μM $NaHCO_3$), and a 0.1 M $H_2SO_4$ regenerant.

## 2.5. Solid-State Analysis

The elemental composition of Puromet MTS9140 and the oxidation state of adsorbed Cu was determined via X-ray photoelectron spectroscopy (XPS) using a 'Kratos AXIS Supra' instrument and monochromated Al source. A small sample of Cu-loaded resin was homogenised in a clean pestle and mortar using a small amount of deionized water to form a paste. This was gently dried overnight at 50 °C to produce a fine powder of ground resin and was submitted to the Sheffield Surface Analysis Centre, where a subsample was pressed into indium foil prior to analysis. Survey scans were carried out between 1200–0 eV energy resolution and one 300 s sweep. High resolution C 1*s*, Cu 2*p*, and Cu LMM scans were collected at their appropriate energy ranges at 0.1 eV energy resolution, with a 300 s sweep for C and three 300 s sweeps for the Cu 2*p* and Cu LMM scans.

## 2.6. Breakthrough Modelling

Ion breakthrough was analysed using multiple breakthrough models commonly applied to ion exchange data; the modified dose–response (MDR), Bohart–Adams, Thomas, and Yoon–Nelson models. It is important to note is that the models were not necessarily intended to describe a solid-liquid ion exchange extraction process at the time they were developed, and as such the calculated values may not accurately describe experimental reality [19]. However, given the widespread use of such models in the field, this remains the only way to consistently compare new experimental data with existing literature, and so

this approach is justified. Model fitting was performed for individual metal breakthrough in OriginPro 2020b software using non-linear regression analysis.

The MDR model is given in Equation (1) [20], where $F_t$ is the cumulative flow-through (mL) at a given time, and $a$ and $b$ are model constants. From evaluating the MDR model, the maximum column loading capacity for each metal ($Q_o$) can also be derived using Equation (2), where $m$ is the mass of resin used (g).

Equation (1). Modified dose response model.

$$\frac{C_t}{C_o} = 1 - \frac{1}{1 + \left(\frac{F_t}{b}\right)^a} \tag{1}$$

Equation (2). Calculation of *Qo* from MDR constant *b*.

$$Q_o = \frac{b \, C_o}{m} \tag{2}$$

The Bohart–Adams model is given in Equation (3), where it is assumed that the rate of adsorption is dependent on both the concentration of sorbing species in solution and on the remaining capacity of the adsorbent. While originally developed for describing a gas-charcoal adsorption system [21,22] the model can also be applied to a solid phase extraction system from a solution phase. In Equation (3), $K_a$ is the Bohart–Adams adsorption rate constant (L mg$^{-1}$ min$^{-1}$), $W$ is the column adsorption capacity (mg/g), and $F$ represents the volumetric flow rate (L/hour).

Equation (3). Bohart–Adams model.

$$\frac{C_t}{C_o} = \left(\frac{e^{K_a C_o t}}{e^{K_a C_o t} + e^{K_a \left(\frac{W}{F}\right)} - 1}\right) \tag{3}$$

The Thomas model (Equation (4), where $K_t$ is the model constant (L min$^{-1}$ mg$^{-1}$) is also commonly applied to ion exchange breakthrough data and was originally developed to describe adsorption to a zeolite bed [23]. A high goodness-of-fit to this model would suggest that uptake is governed by mass transfer at the resin-solution interface [24].

Equation (4). Thomas model.

$$\frac{C_t}{C_o} = \frac{1}{1 + e^{\left(\frac{K_t Q_o m}{F} - K_t C_o t\right)}} \tag{4}$$

The final model fitted to breakthrough data is the Yoon–Nelson model, originally developed for describing the adsorption of gases to solid adsorbents, is presented in Equation (5) where $K_{yn}$ is the model constant (min$^{-1}$), and $t_{50}$ is the predicted time for 50% breakthrough to be–a useful parameter for understanding column operating times.

Equation (5). Yoon–Nelson model.

$$\frac{C_t}{C_o} = \frac{1}{1 + e^{K_{yn}(t_{50} - t)}} \tag{5}$$

## 3. Results and Discussion

### 3.1. Static (Batch) Extraction

When contacted with the PLS under batch conditions, Puromet MTS9140 exhibited great selectivity towards Cu across all studied pH conditions (Figure 1), with extraction remaining above 92% irrespective of increased proton concentration. The extraction of other metals present in the PLS was very low, with almost no change in extraction percentage with respect to equilibrium pH.

Increased sulphate concentration in the PLS had no observable effect on the extraction of any metal species by S914, with the high capacity for Cu removal and low co-removal of other metals reported under all sulphate concentration conditions (Figure 2), indicating

a high resilience to increased ionic strength. Given that this resin was not developed by the manufacturer for Cu separations, and that true selectivity towards a single ion would be highly advantageous for treatment of mixed metal waste streams, further exploration into the surface binding of Cu was explored for this resin by using X-ray Photoelectron Spectroscopy (XPS) analysis to better understand the extraction process.

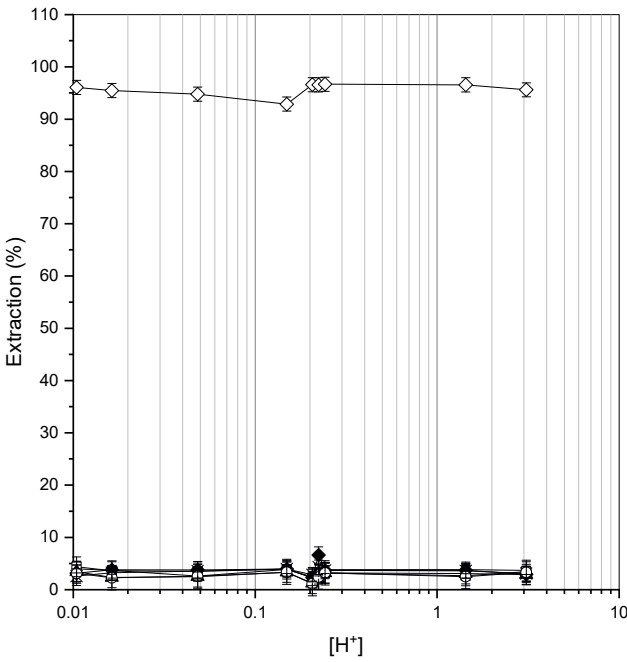

**Figure 1.** Extraction percentage of metal ions as a function of acid concentration on MTS9140. Al(III) = +, Co(II) = □, Cu(II) = ◇, Fe(III) = ◆, Ni(II) = ▲, Mn(II)= ✕, Zn(II) = ○.

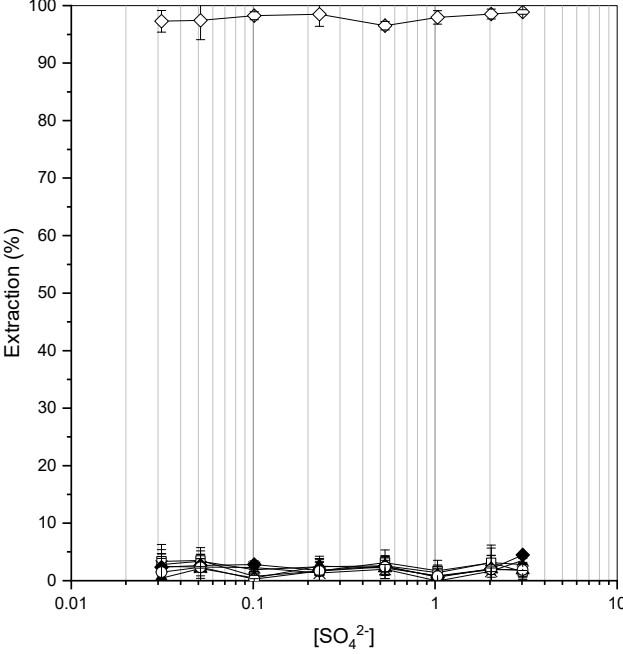

**Figure 2.** Extraction percentage of metal ions as a function of ammonium sulphate concentration on MTS9140 at 0.027 M H⁺. Al(III) = +, Co(II) = □, Cu(II) = ◇, Fe(III) = ◆, Ni(II) = ▲, Mn(II)= ✕, Zn(II) = ○.

The XPS survey scan was used to determine surface elemental composition of powdered sample of Cu-loaded S914, given as atomic % (At%) in Figure 3. As can be expected for a resin bead with a polystyrene-DVB backbone, the most abundant element quantified was C (63.77 At%). In decreasing abundance, the remaining composition was determined to be O (16.98 At%), N (7.62 At%), Si (7.10 At%), S (4.28 At%), and Cu (0.25 At%) (Figure 3). All elemental peaks were clearly detected, as evidenced by the low 'full width at half maximum' (FWHM) values reported in the table within Figure 3.

The determined binding energy of N (400 eV) is consistent with nitrogen within an organic matrix, i.e., the thiourea functional group within the resin matrix. While the presence of N was expected given the functionality of the resin, the high Si and O concentrations were not expected. The binding energies of Si $2p$ and O $1s$ peaks in Figure 3 were consistent with the presence of silica ($SiO_2$), yet the percentage composition of oxygen indicates that silica was not the only oxygen source present in the sample. Close examination of the S $2s$ peak revealed two components; a peak consistent with thiourea (75% of S detected), and a smaller peak (25% of S detected) attributed to the presence of sulphate–presumably residual sulphate from the PLS during loading.

Considering that the resin was loaded with excess Cu in solution, the Cu surface concentration was relatively small (0.25 At%). This was likely due to a diluting effect of the bulk resin matrix (PS-DVB), which could overshadow the presence of functional groups. The low concentration of thiourea relative to the bulk matrix meant that the photoelectrons emitted from Cu atoms (associated with thiourea groups) originated too deep within the sample to escape without further collision and were less-well detected.

In addition to the full survey scan to determine bulk elemental composition, a more detailed scan of the Cu $2p$ environment was performed to investigate the speciation of Cu when loaded to S914. Figure 4 indicates a clear symmetrical peak at 932.82 eV, peaking at approximately 214 counts per second (CPS). A second, less well-defined peak was also observed in the region of 952.5 eV, with a magnitude of 190 CPS. While the emission baseline appeared noisy during the Cu $2p$ scan, this is a direct result of the relatively low At% of Cu in the Cu-loaded resin sample (Figure 3).

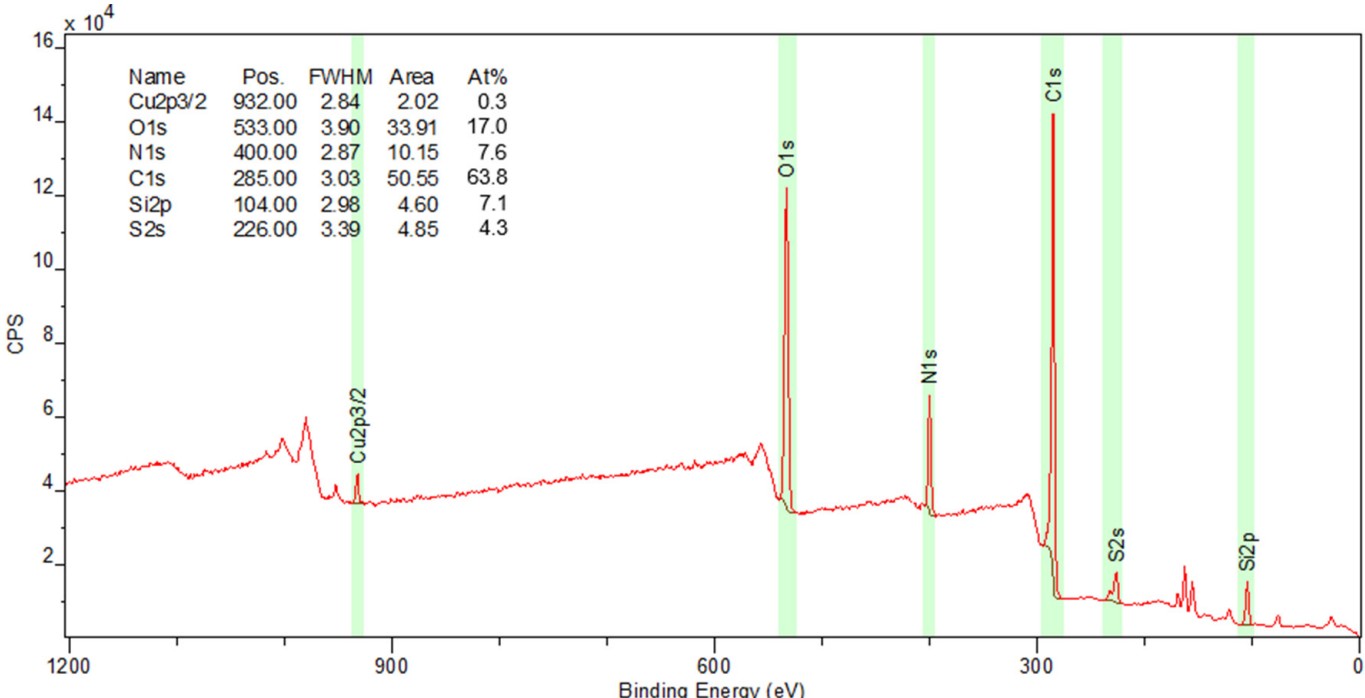

**Figure 3.** XPS survey scan for determination of elemental composition of Cu-loaded MTS9140 (CPS = counts per second).

The lack of Cu satellite peaks in the 940–945 eV region eliminated the possibility for Cu(II) being present within the sample [25], instead the binding energy of the Cu 2*p* peak was consistent with either Cu metal or Cu(I) (933 eV for both species), and the secondary peak at 952.5 eV coupled with the absence of satellite peaks provides further evidence for this [26].

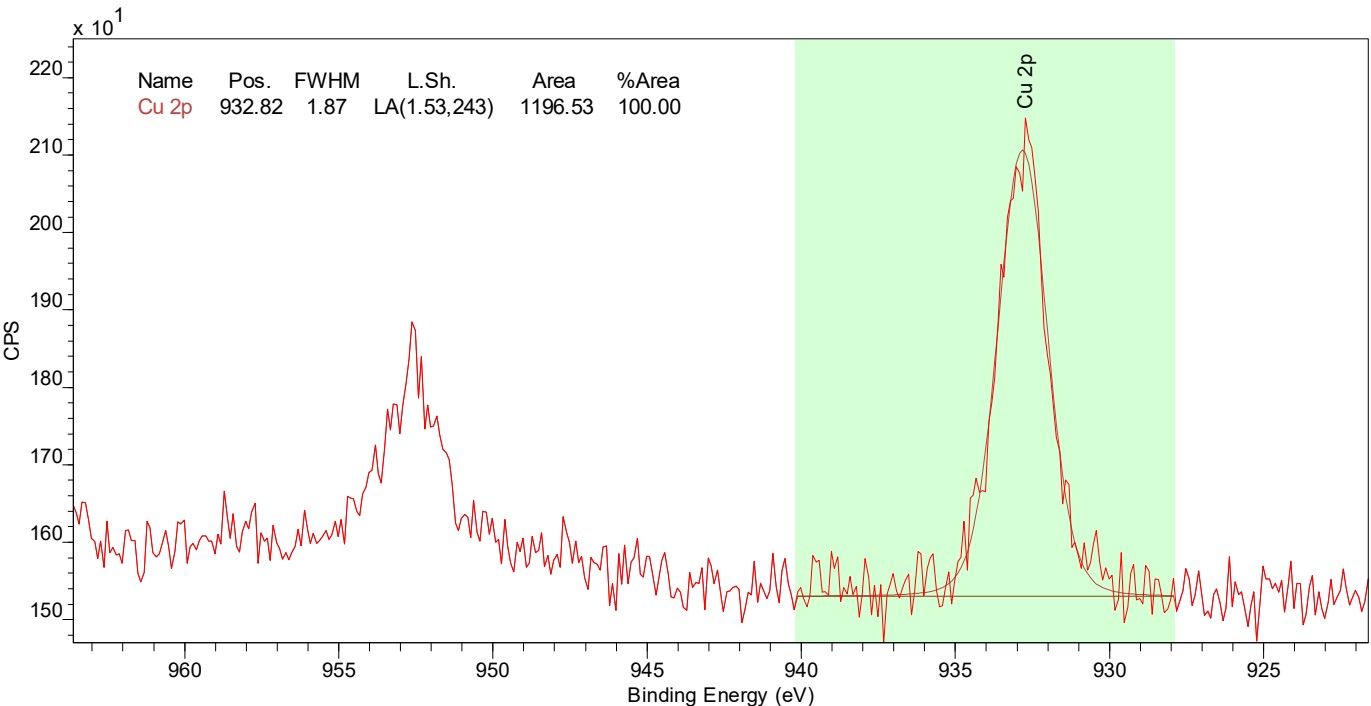

**Figure 4.** XPS Cu 2*p* scan for Cu-loaded MTS9140 (CPS = counts per second).

For Cu to be present on the resin surface as either Cu(I) or Cu metal, it follows that the Cu(II) present in solution must undergo reduction, with the functional group of the resin in turn being oxidised. Such redox behaviour between Cu(II) and solutions of thiourea has been previously observed [27], whereby Cu(II) is immediately bonded with thiourea which is in turn quickly oxidised by Cu(II) ions. The resulting Cu(I) ion produced as a result of this redox reaction is then complexed by thiourea to form a stable Cu(I)-thiourea complex, the form of which may involve one, two, or three thiourea ligands [28]. Given the reported interactions of Cu(II) ions with thiourea ligands in solutions, and given that Cu metal would be expected to give a sharper and more asymmetric peak than observed in Figure 4 [29], it is proposed that S914 removes Cu from solution through reduction to cuprous Cu(I).

### 3.2. Fixed-Bed Adsorption

Under dynamic operation, S914 continued to exhibit exclusive Cu selectivity and extraction from the PLS as evidenced by the almost immediate breakthrough of all other ions from the column, which reached complete breakthrough within the first five BV throughput (Figure 5). Numerical modelling for these metals indicated very low loading capacities for these ions (2.02–2.15 mg/g, MDR, Table 2), but considering the speed at which these metals broke through and considering the lack of displacement following complete breakthrough, this is likely an overestimation of loading capacity.

Cu breakthrough began to occur at around 5 BV throughput and gradually increased, following a slightly sigmoidal pattern, until reaching a concentration ration of 0.92 at BV 80 (Figure 5). While the breakthrough curve was not sharp, Cu extraction appeared unaffected by other metal ions in solution and was best-defined by the MDR model, which indicated a loading capacity of 19.84 mg/g.

Changing solution flow rate to 5 BV/h had little effect on the uptake of Co, Fe, Mn, Ni, and Zn, which passed through the column with little-to-no interaction with the thiourea-functionalised resin (Figure 6). For Cu, which was exclusively extracted by S914, the breakthrough profile appeared sharper than it did when loaded at 10 BV/h, which was confirmed by an associated increase in the MDR model constant 'a' (Table 3). Complete Cu breakthrough was also achieved earlier at the 5 BV/h flow rate, where Cu concentration ratio reached 1 at 68 BV throughput and corresponded to a $Q_o$ value of 20.68–22.77 mg/g according to the breakthrough modelling (MDR and Thomas model had equal goodness-of-fit, $R^2 = 0.998$).

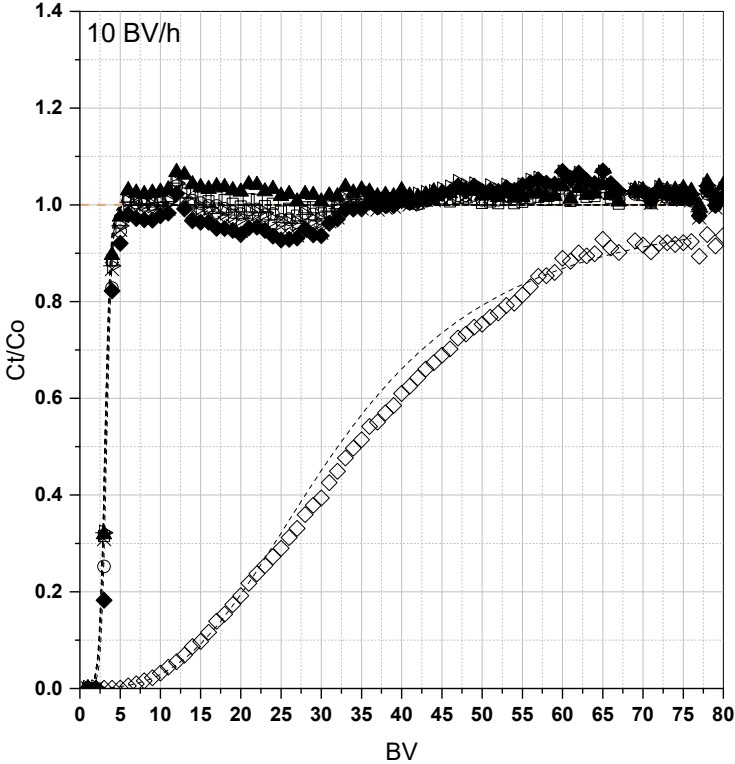

**Figure 5.** Breakthrough curves of metals from PLS pumped through MTS9140 at 10 BV/h (pH 1.56). Al = ▷, Co = □, Cu = ◇, Fe = ◆, Mn = ×, Ni = ▲, Zn = ○. Dotted lines = best-fitting breakthrough model (see Table 2).

**Table 2.** Breakthrough model parameters for MTS9140 at 10 BV/hour flow rate.

|  | Modified Dose Response | | | | Bohart–Adams | | | Thomas | | | Yoon–Nelson | | |
|---|---|---|---|---|---|---|---|---|---|---|---|---|---|
|  | *a* | *b* | $Q_o$ | $R^2$ | $K_a$ | *W* | $R^2$ | $K_t$ | $Q_o$ | $R^2$ | $K_{yn}$ | $t_{50}$ | $R^2$ |
| Al | 9.15 | 15.45 | 2.10 | *0.996* | 0.14 | 3.27 | *0.996* | 0.14 | 2.12 | *0.996* | 0.49 | 18.22 | *0.998* |
| Co | 9.64 | 15.44 | 2.15 | *0.999* | 0.14 | 3.34 | *0.999* | 0.14 | 2.17 | *0.999* | 0.52 | 18.20 | *0.996* |
| Cu | 3.01 | 160.26 | 19.84 | *0.998* | 0.005 | 33.07 | *0.989* | 0.005 | 21.20 | *0.989* | 0.02 | 199.98 | *0.989* |
| Fe | 9.97 | 16.48 | 2.13 | *0.980* | 0.16 | 3.31 | *0.979* | 0.16 | 2.15 | *0.979* | 0.53 | 19.44 | *0.995* |
| Mn | 9.06 | 15.57 | 2.02 | *0.986* | 0.15 | 3.14 | *0.985* | 0.15 | 2.04 | *0.985* | 0.49 | 18.36 | *0.997* |
| Ni | 10.07 | 15.37 | 2.10 | *0.999* | 0.15 | 3.26 | *0.999* | 0.15 | 2.11 | *0.999* | 0.54 | 18.11 | *0.997* |
| Zn | 10.07 | 15.37 | 2.01 | *0.999* | 0.14 | 3.27 | *0.997* | 0.14 | 2.12 | *0.997* | 0.48 | 18.96 | *0.999* |

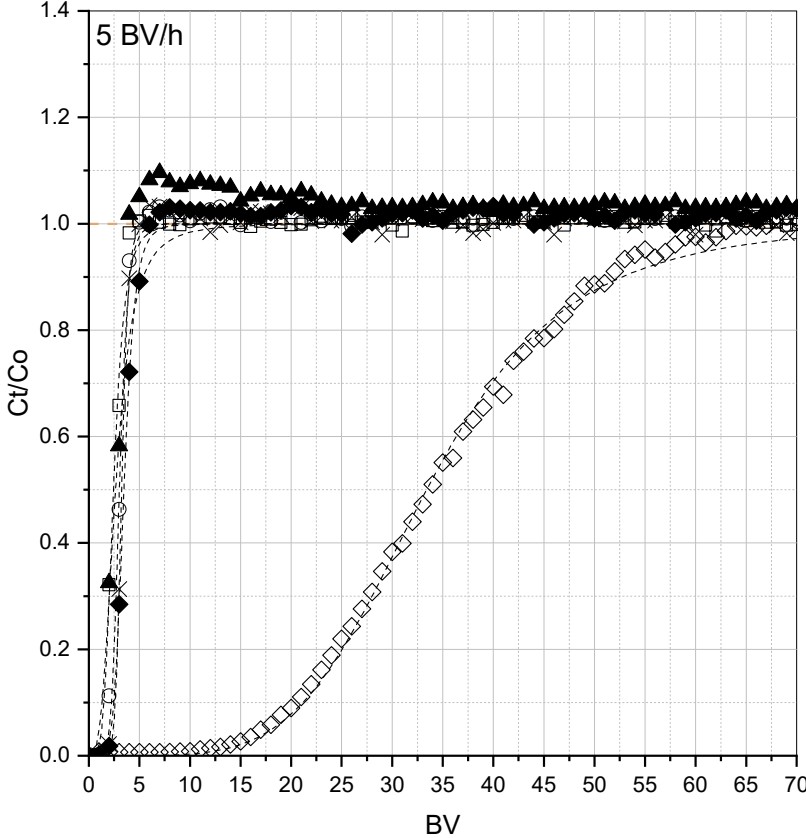

**Figure 6.** Breakthrough curves of metals from PLS pumped through MTS9140 at 5 BV/h (pH 1.56). Co = □, Cu = ◇, Fe = ◆, Mn = ×, Ni = ▲, Zn = ○. D otted lines = best-fitting breakthrough model (see Table 3).

**Table 3.** Breakthrough model parameters for MTS9140 at 5 BV/hour flow rate.

|      | Modified Dose Response | | | | Bohart-Adams | | | Thomas | | | Yoon-Nelson | | |
|      | a | b | $Q_o$ | $R^2$ | $K_a$ | W | $R^2$ | $K_t$ | $Q_o$ | $R^2$ | $K_{yn}$ | $t_{50}$ | $R^2$ |
|------|------|--------|-------|-------|-------|-------|-------|-------|-------|-------|----------|----------|-------|
| Co | 4.49 | 12.15 | 1.69 | *0.994* | 0.05 | 2.72 | *0.995* | 0.05 | 1.76 | *0.995* | 0.18 | 26.52 | *0.992* |
| Cu | 4.81 | 167.02 | 20.68 | *0.998* | 0.004 | 35.13 | *0.998* | 0.004 | 22.77 | *0.998* | 0.01 | 360.37 | *0.998* |
| Fe | 6.45 | 17.32 | 2.23 | *0.999* | 0.05 | 3.50 | *0.997* | 0.05 | 2.27 | *0.997* | 0.18 | 36.91 | *0.996* |
| Mn | 10.29 | 16.18 | 2.10 | *0.999* | 0.09 | 3.26 | *0.999* | 0.09 | 2.12 | *0.999* | 0.28 | 34.22 | *0.998* |
| Ni | 3.09 | 13.20 | 1.80 | *0.987* | 0.04 | 2.85 | *0.956* | 0.04 | 1.84 | *0.956* | 0.17 | 27.37 | *0.992* |
| Zn | 7.35 | 15.07 | 1.87 | *0.995* | 0.07 | 3.05 | *0.999* | 0.07 | 1.98 | *0.999* | 0.23 | 31.79 | *0.998* |

When reduced to 2 BV/h, a further sharpening of the breakthrough of ions was observed. For all metals other than Cu, almost immediate complete breakthrough occurred, after which concentration ratios plateaued at values between 1 and 1.1 (Figure 7). It is important to note that a decision was made to increase sampling resolution during the 2 BV/h flow rate study to increase the number of data points during the frontal portion of the breakthrough curve. Given the limited space for sample collection in fraction collectors, this resulted in only the first 40 BV being sampled. Nevertheless, sufficient Cu breakthrough was captured to allow effective modelling and comparison to other flow rates. Cu was first detected in effluent solutions beyond 25 BV throughput (Figure 7), much later than observed for 10 and 5 BV/h (BV 5 and 10, respectively).

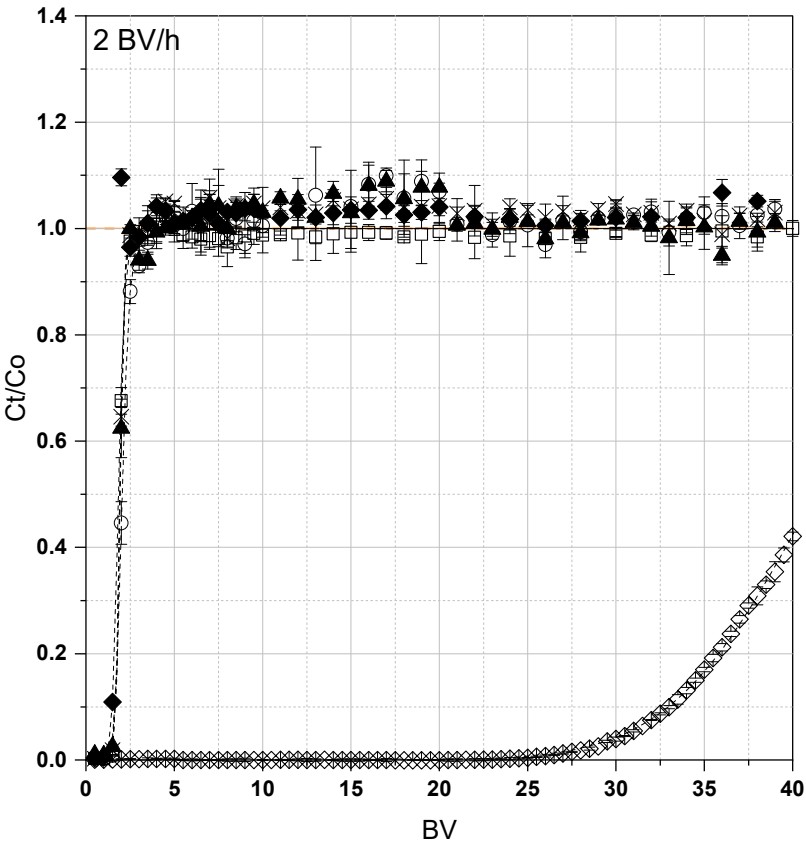

**Figure 7.** Breakthrough curves of metals from PLS pumped through MTS9140 at 2 BV/h (pH 1.56). Co = □, Cu = ◇, Fe = ◆, Mn = ×, Ni = ▲, Zn = ○. Dotted lines = best-fitting breakthrough model (see Table 4).

**Table 4.** Breakthrough model parameters for MTS9140 at 2 BV/hour flow rate.

|    | Modified Dose Response | | | | Bohart-Adams | | | Thomas | | | Yoon-Nelson | | |
|----|------|-------|-------|-------|-------|-------|-------|-------|-------|-------|----------|--------|-------|
|    | a     | b      | $Q_o$  | $R^2$   | $K_a$   | W      | $R^2$   | $K_t$   | $Q_o$  | $R^2$   | $K_{yn}$  | $t_{50}$ | $R^2$   |
| Co | 16.67 | 9.56   | 1.46   | *0.996* | 0.09   | 2.25   | *0.997* | 0.09   | 1.46   | *0.997* | 0.333    | 57.52   | *0.999* |
| Cu | 9.74  | 206.45 | 32.04  | *0.999* | 0.002  | 48.75  | *0.997* | 0.002  | 31.66  | *0.997* | 0.01     | 1223.9  | *0.997* |
| Fe | 10.46 | 9.16   | 0.94   | *0.999* | 0.07   | 1.48   | *0.999* | 0.07   | 0.96   | *0.999* | NA       | NA      | *NA*    |
| Mn | 16.14 | 9.63   | 1.08   | *0.999* | 0.10   | 1.66   | *0.999* | 0.10   | 1.08   | *0.999* | 0.30     | 57.72   | *0.999* |
| Ni | 16.57 | 9.69   | 1.18   | *0.999* | 0.09   | 1.81   | *0.999* | 0.09   | 1.18   | *0.999* | 0.29     | 57.94   | *0.999* |
| Zn | 9.97  | 10.25  | 1.13   | *0.996* | 0.06   | 1.73   | *0.996* | 0.06   | 1.13   | *0.996* | 0.17     | 61.52   | *0.996* |

As was the case for batch pH and sulphate concentration screening, no published data exists studying the fixed-bed adsorption of metals using Puromet MTS9140. Under dynamic operation, MTS9140 displayed the same Cu-selective extraction properties as under previous batch experiments. All other metals present in the PLS reached complete breakthrough within the first 5 BV and exhibited very little overshoot ($C_t/C_o > 1$). The brief period of overshoot observed in Figure 6, which increased slightly as flow rate decreased (Figure 7), indicating that, upon entry to the column, metals were momentarily adsorbed but rapidly displaced as Cu began to load. The fact that all other metals reach complete breakthrough long before Cu indicated that it was not competition for remaining active sites that led to the displacement of metals, but instead that Cu was able to replace metal counter ions immediately upon contact and before resin saturation.

All breakthrough models were able to describe metal loading to MTS9140 with a high degree of accuracy (Tables 2–4), with the MDR model generally outperforming other models for Cu. Results of MDR modelling revealed Cu operating capacities of 19.84 mg/g

at 10 BV/h, 20.68 mg/g at 5 BV/h, and 32.04 mg/g at 2 BV/h. While Cu capacity at the high and intermediate flow rates were comparable, a notable increase in capacity was observed at the lowest flow rate. This was likely the result of a more effective extraction at the low flow rate, evidenced by delayed breakthrough point and longer column half-lives calculated by the Yoon–Nelson model (20.4 min) when compared to 5 BV/h (6 min) and 10 BV/h (3.3 min) operation.

### 3.3. Resin Elution

The proposed reductive extraction mechanism of Puromet MTS9140 suggested an oxidative eluent to be applied to liberate Cu(I) as Cu(II) from the resin. While no literature exists on the elution of Cu from Puromet MTS9140 specifically, a limited number of articles do exist that explore the elution of Cu from non-commercial dual-functionalized resins containing, among other groups, thiourea functionality. One such paper reports effective batch Cu recovery from a thiourea/acyl bifunctional resin using concentrated nitric acid (equivalent to 3.0 M) as an eluent [30], and was initially explored (see Supplementary Material).

Based on initial experimentation, it was deemed that the use of concentrated nitric acid as an eluent for MTS9140 would be unsuitable for two main factors; (1) the observed formation of nitrous gases could hinder process scale-up, and (2) the safety concerns of handling large volumes of highly concentrated nitric acid after process scale-up. However, given the presence of Cu(I) on the thiourea resin surface, an oxidative elution approach remained a possible avenue of exploration, and so sodium chlorate ($NaClO_3$) was chosen for further elution studies, given the higher reduction potential of chlorate (1.152–1.451 V) over nitrate (0.803–0.934 V) in acidic media [31]. In addition to acting as a stronger oxidising agent, the use of $NaClO_3$ over $HNO_3$ allowed elution to be performed under less-acidic conditions.

The elution profile of Cu from S914 using a 0.5 M solution of $NaClO_3$ at pH 2 is presented in Figure 8. The concentration of Cu in effluent solutions increased sharply beyond 4 BV throughput, reaching a maximum concentration of 511 mg/L at 20 mL (~14 BV) throughput. Following peak maximum, Cu concentration exhibited a steep decline, which gradually levelled out to below 10 mg/L by the end of the experimental run; allowing the complete profile to be captured. Integration of the area below the curve (Table 5) revealed effective Cu elution by this eluent, with an overall Cu recovery percentage of 78.91%.

**Table 5.** Details of Cu elution investigations using $NaClO_3$ (FWHM provided for comparison of peak widths).

| [$NaClO_3$] | Cu Loaded (mg/mL) | Bed Volume (mL) | Total Cu on Bed (mg) | Cu Recovered (mg) | FWHM (mL) | Recovery Efficiency (%) |
|---|---|---|---|---|---|---|
| 0.5 M | 8.64 | 1.4 | 12.10 | 9.55 | 16.2 | 78.91 |
| 1.0 M | 8.36 | 1.4 | 11.70 | 9.58 | 13.2 | 81.86 |

Doubling the concentration of sodium chlorate had the effect of increasing the maximum Cu concentration to 612 mg/L (Figure 9). Despite the higher Cu concentration during peak maximum, the width of the elution peak (FWHM) was smaller than that in Figure 8, resulting in a recovery efficiency of 81.86%; fairly similar to the Cu recovery using 0.5 M $NaClO_3$ (Table 5). Following elution using 1 M $NaClO_3$, a slight but notable colour change was observed, with the 1M-contacted S914 taking on a grey hue when compared to the 0.5 M-contacted resin. It is theorised that this grey colouration was the result of copper oxide formation on the surface of the resin bead.

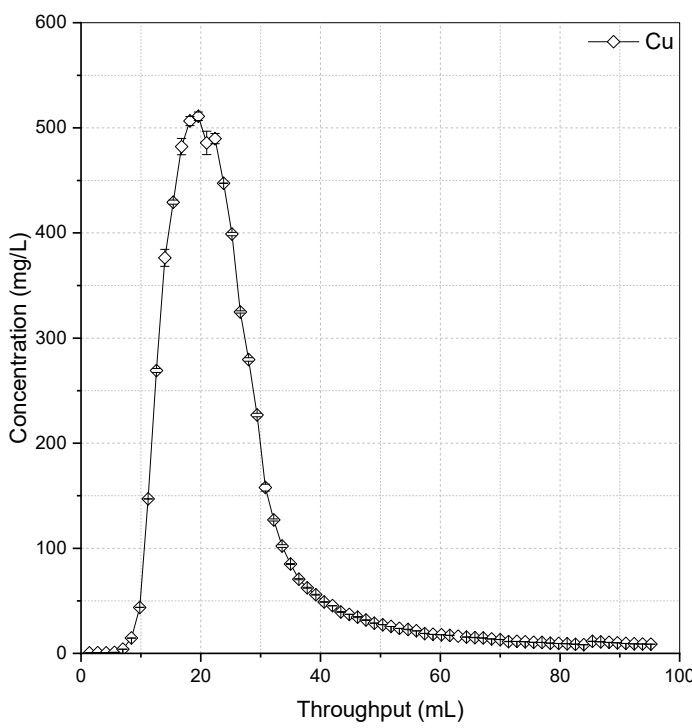

**Figure 8.** Elution of Cu from MTS9140 using 0.5 M NaClO3 at pH 2 (HCl matrix, 2 BV/h).

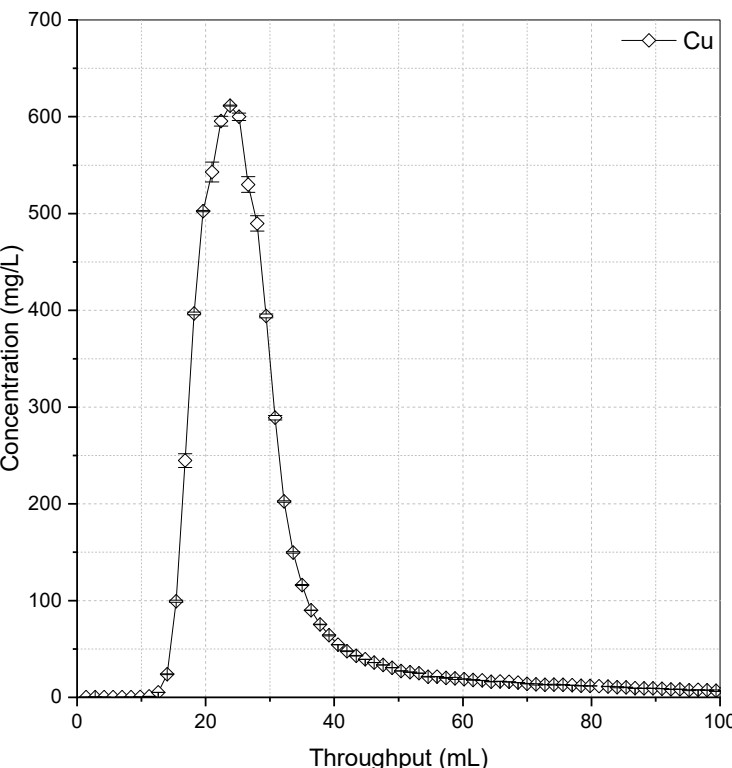

**Figure 9.** Elution of Cu from MTS9140 using 1 M NaClO$_3$ at pH 2 (HCl matrix, 2 BV/h).

No gas formation was observed during chlorate elution, which was also evidenced by the more regular shape of elution profiles when compared to those obtained during HNO$_3$ elution (see Supplementary Information). Cu elution profiles were fairly symmetrical with a steep frontal curve and slightly longer tail-end. This curve profile is often observed where

effective elution is achieved [32–34], and implies a favourable exchange (or removal) of the target ion in favour of the eluent counter-ion.

### 3.4. Resin Reusability

To determine the reusability of MTS9140 for the selective extraction of Cu, repeated loading, elution, and preconditioning cycles were performed. The 0.5 M NaClO$_3$ eluent was used given its effectiveness for Cu recovery. While the 1 M eluent did have a marginally higher recovery efficiency, the lack of resin discoloration indicated that 0.5 M NaClO$_3$ was a better option for investigating reusability. The resulting breakthrough curves are presented in Figure 10.

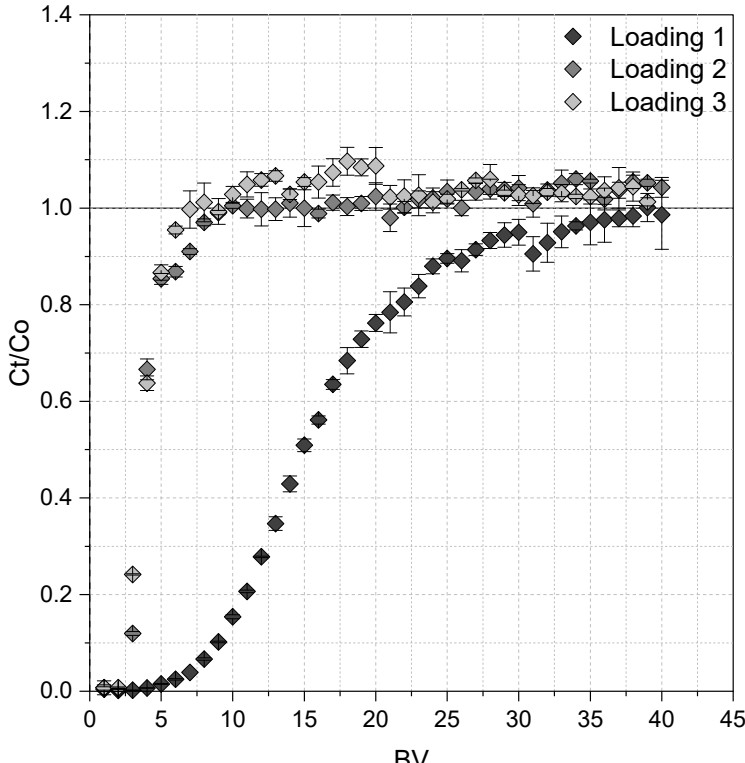

**Figure 10.** Breakthrough curves of Cu from MTS9140 over repeated loading cycles (1.4 mL BV, 5 BV/h; 400 mg/L Cu, pH 1.55).

During the first loading cycle, typical Cu adsorption behaviour was observed, with complete breakthrough encountered at 40 BV throughput (Figure 10). Integration of the area above the breakthrough curve (with an upper boundary of influent PLS concentration) indicated the adsorption of 7.95 mg Cu during the first loading cycle. Following Cu elution, the resin bed was loaded again using the 400 mg/L Cu solution, but exhibited very low adsorption of Cu, with only 1.55 mg of Cu removed before complete breakthrough occurred during both the second and third loading cycles (Table 6).

Comparison of elution curves using 0.5 M NaClO$_3$ (Figure 11) revealed that for the first cycle, where 7.95 mg Cu was loaded to the resin bed, 6.79 mg was recovered, equating to a recovery efficiency of 85.34% (Table 6). The elution profile generated increased sharply, reaching a maximum Cu concentration of 418 mg/L after 0.028 L, before decreasing in an almost-symmetrical fashion, indicating effective desorption.

The considerably lower extent of Cu extraction during the second and third loading cycles (Figure 10) was reflected in the elution profiles for the respective cycles (Figure 11). However, for elution cycle two, the recovery efficiency was consistent with recovery during cycle one (88.46 and 85.34, respectively; Table 6). While the extent of Cu loading between

cycle one and two was similar, the mass of Cu eluted differed greatly, with a decline in recovery efficiency from 88.46% to only 46.27% during elution cycle three (Table 6).

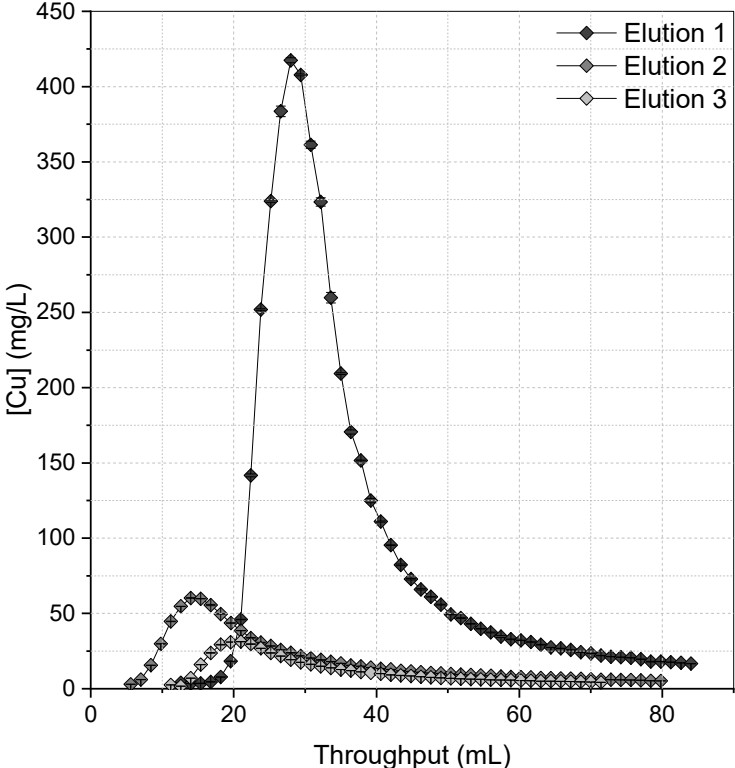

**Figure 11.** Elution of Cu from MTS9140 using 0.5 M NaClO$_3$ (pH 2, HCl matrix, 2 BV/h) over repeated elution cycles.

**Table 6.** Masses of Cu loaded and eluted from MTS9140 over multiple operational cycles.

| Cycle | Loading (mg) | Elution (mg) | Recovery (%) |
|-------|--------------|--------------|--------------|
| 1 | 7.95 | 6.79 | 85.34 |
| 2 | 1.55 | 1.37 | 88.46 |
| 3 | 1.55 | 0.72 | 46.27 |

The lower mass of Cu recovered from S914 during the third elution cycle could potentially be explained by the progressive loss of resin functionality during elution cycles. It could be assumed that as the eluent contacts the exterior of resin beads before diffusing through the resin pores, the functional groups at the surface of each bead would be the first to be degraded, explaining the loss of capacity between cycle one and two. Assuming this 'shrinking-core' hypothesis for loss of functionality is true, it holds that during the later elution cycles, the remaining functionality is located deeper within the resin beads and therefore less accessible to the eluent. While the ionic radii of Cu(I) (0.46–0.77 Å) is substantially smaller than that of ClO$_3^-$ (1.71 Å), it is unlikely that Cu residing in pores inaccessible to ClO$_{3-}$ was responsible for this observation given the flexibility of gelled polymers [18]. Instead, this is more likely a result of kinetic limitations inherent to column operation, i.e., the limited residence time within the column is hindering sufficient mass transfer between the bulk eluent and resin, and in doing so reduces eluent efficiency [35]. It is expected that such effects are amplified when coupled with functionality degradation on resin outer surfaces.

### 3.5. Functionality Degradation

It is evident that while Cu can successfully and efficiently be recovered from MTS9140, a cuprous oxidation approach to elution is unsuitable for maintaining the functionality of the resin for reuse. To better understand this degradation of MTS9140, Cu elution using 0.5 M NaClO$_3$ was repeated on a Cu-loaded column, with effluent bed volumes being sampled and analysed by IC.

During elution of Cu, an increase in pH was observed in effluent solutions, peaking at pH 3.13 at 7 BV throughput (Figure 12); an increase of 1.18 pH units from the native pH of the eluent used (pH 1.95). A lag of 5 BV was observed until peak Cu elution, which occurred after 12 BV and reached a concentration of 604 mg/L.

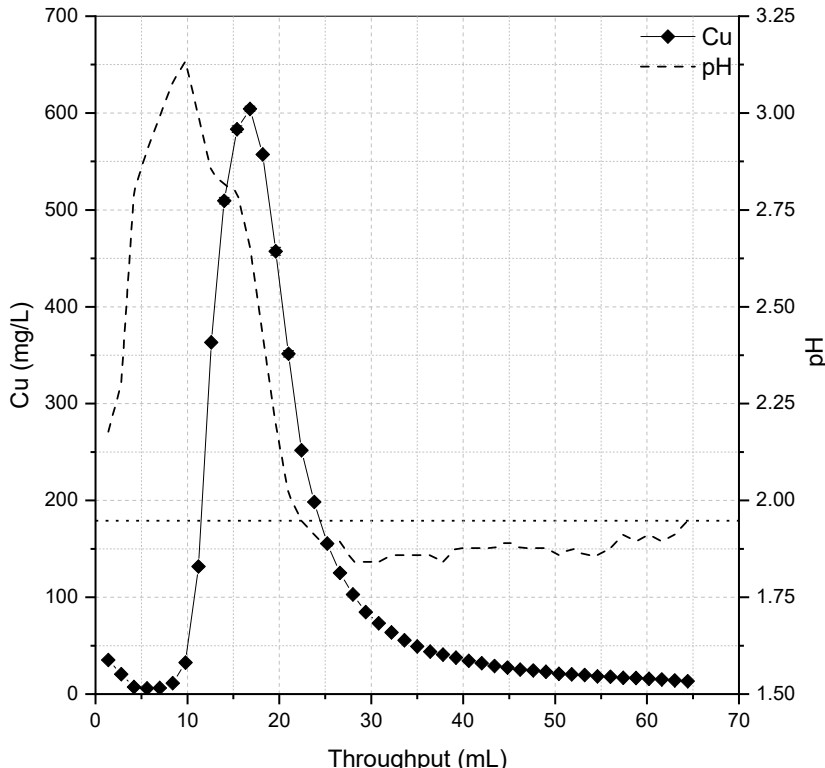

**Figure 12.** Cu concentration and pH of effluent solutions during elution of Cu from MTS9140 at 2 BV/h using 0.5 M NaClO$_3$ (pH 1.95, HCl media, dotted horizontal line represents pH of eluent).

Given that the chlorate ion is fundamental to the oxidation of the Cu(I) centre, it was theorised that a peak in chloride would occur alongside the Cu elution peak, and so this was analysed for by IC, as well as sulphate concentrations. The concentrations of Cl$^-$ and SO$_4{}^{2-}$ are presented alongside the Cu elution profile in Figure 13. A peak in chloride concentration was observed simultaneously with peak Cu elution, with a maximum concentration of 0.02 M Cl$^-$ (714 mg/L). A substantial peak in sulphate concentration (maximum 0.085 M SO$_4{}^{2-}$ (5646 mg/L)) was also detected, occurring prior to the peak in Cu and Cl, and before the period of increased pH.

The peak in Cl$^-$ is not unexpected, and the parallel occurrence of Cu and Cl$^-$ elution confirmed that the method of Cu liberation is a redox interaction between ClO$_{3-}$ and Cu(I), whereby ClO$_{3-}$ is reduced to Cl$^-$ and Cu(I) oxidised to Cu(II). However, the large spike in sulphate concentration is particularly significant given that no sulphate was introduced to the system during the elution cycle. While the PLS used for loading did contain sulphate, a thorough rinse cycle using 18 MΩ deionised water was performed prior to elution such that residual sulphate concentration was below 0.002 M prior to the start of the sulphate peak in Figure 13.

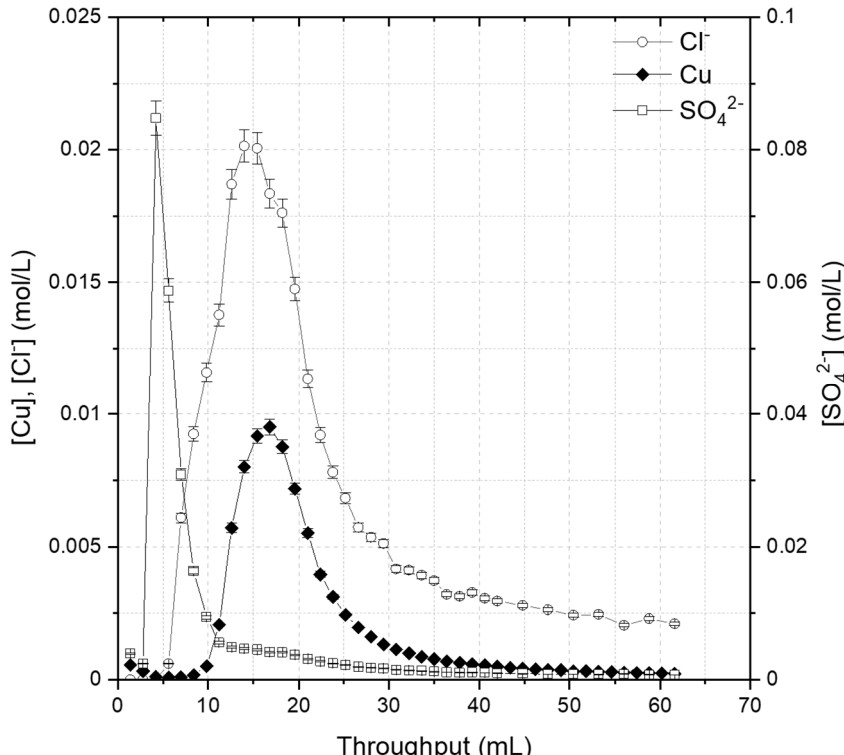

**Figure 13.** Concentrations of Cu, Cl⁻, and SO₄²⁻ in effluent solutions during elution of S914 at 2 BV/h using 0.5 M NaClO₃ (pH 1.95, HCl media, same Cu elution profile as presented in Figure 12).

It is important to note that anion concentrations presented in Figure 13 have been corrected to consider the concentrations in both the calibration blank and eluent solution, such that the presented data represents only additional anions introduced to the system. Therefore, the sulphate peak detected in effluent solutions could have only originated as a result of resin degradation, and it is likely that this degradation is responsible for the diminished capacity observed during cyclic adsorption (Table 6). While it is possible that other processes may play a part in this reduced performance, such as the possible presence of a passive CuO film during secondary loading cycles, the spike in sulphate occurring prior to Cu elution (Figure 13) indicates that functionality degradation is the key driver of this.

While not specifically recorded for chlorate, the oxidation of thiourea compounds by other halogenic oxidants such as bromate, chlorite, and iodate has been shown to form sulphate through substitution of sulphur for oxygen on the thiourea group, and subsequent sulphur oxidation [36]. It is therefore expected that oxidation by chlorate is also able to produce such by-products during degradation of thiourea functional groups.

## 4. Conclusions

The extractive behaviour of a thiourea-functionalized resin, Puromet MTS9140, from an acidic mixed-metal system was studied under static and dynamic conditions, where a highly selective behaviour towards Cu extraction was observed. The mechanism of extraction was elucidated to be via reduction of Cu(II) to Cu(I) by the thiourea functionality, and an efficient oxidative elution approach was used to recover a concentrated Cu product stream under ambient conditions, albeit at expense of resin reusability. This holds potential benefit in the separation of Cu from other base metals and will aid in the development of a cost-effective Cu recovery process for a variety of existing primary and secondary sources, including emerging Cu sources in waste reprocessing.

Copper-based catalysts are widely used in chemical industries, and in addition to the mining sector, this resin may prove useful in sectors such as the pharmaceutical industry in their recent move away from using precious metal catalysts in favour of more environmentally-

benign copper catalysts [37–39]. Given the tight regulations of Cu content in end-products, this target-specific resin may be of particular interest for this application also.

The issue of resin reusability brings to light opportunities in further research. The application of Puromet MTS9140 for Cu recovery from real waste and/or ore leachates would be beneficial for determining industrial applications of this resin and would allow for further optimisation of experimental parameters to suit particular needs, particularly in relation to heterogeneous metal concentrations. Further to this, the exploration of alternative low-cost materials to use as the backbone in thiourea-functionalized adsorbents would be of particular interest to explore further, given the impacts of oxidative Cu recovery on extractive performance. Ongoing research into the functionalization of silica products for metal recovery in other industries (e.g., [40,41]) may offer a solution to the issue of single-use polystyrene-DVB resins; improving overall sustainability in the process through more environmentally-conscious disposal options.

**Supplementary Materials:** The following are available online at https://www.mdpi.com/article/10.3390/eng2040033/s1, Nitric Acid Elution of Cu from Puromet MTS9140; Figure S1. Breakthrough curve of Cu from MTS9140 (5 mL BV, 5 BV/h, 400 mg/L Cu, pH 1.55); Figure S2. Comparison of Cu elution profiles from MTS9140 using 3 M $HNO_3$ at 2 BV/h (D = dynamically-loaded resin, B = batch-loaded resin); Table S1. Details of Cu elution investigations using $HNO_3$ (FWHM provided for comparison of peak widths); Figure S3. Elution of Cu from MTS9140 using 1 M $HNO_3$ at 2 BV/h.

**Author Contributions:** A.L.R.: Conceptualization, Methodology, Formal Analysis, Investigation, Data Curation, Writing—Original Draft, Visualization. C.P.P.: Validation, Resources, Writing—Review & Editing. M.D.O.: Conceptualization, Methodology, Validation, Writing—Review & Editing, Supervision, Funding acquisition. All authors have read and agreed to the published version of the manuscript.

**Funding:** This work was completed as part of a Doctoral Training Partnership PhD program (A.L. Riley) co-funded by the Engineering and Physical Sciences Research Council (EPSRC) and The University of Sheffield.

**Data Availability Statement:** The data presented in this study are available on request from the corresponding author. The data are not publicly available at present.

**Acknowledgments:** The authors would like to acknowledge the members of the Separations and Nuclear Chemical Engineering Research (SNUCER) group at the University of Sheffield who provided support in understanding the results presented in this work. Additionally, Will Mayes of the University of Hull is thanked for provision of ICP-OES analysis for static screening experiments, and Deborah Hammond of the Sheffield Surface Analysis Centre is thanked for XPS analysis. Purolite Ltd. are thanked also for donation of a range of ion exchange resins used as part of wider experimentation.

**Conflicts of Interest:** The authors declare no conflict of interest. The funders had no role in the design of the study; in the collection, analyses, or interpretation of data; in the writing of the manuscript, or in the decision to publish the results.

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
