# Peer review of "Selective Recovery of Copper from a Synthetic Metalliferous Waste Stream Using the Thiourea-Functionalized Ion Exchange Resin Puromet MTS9140"

_2673-4117, doi:10.3390/eng2040033_

Round 1

Reviewer 1 Report

  1. Line 53 needs to be reviewed.
  2. In this manuscript, the atomic concentration of elements (Figure 3) extracted from the XPS data is presented with two digits after the decimal. The authors need to show their XPS results are repeatable. Otherwise, only one digit after the decimal is recommended
  3. The spectroscopic notation should have the orbitals in italics

Author Response

The authors thank the reviewer for their constructive comments on the manuscript, and for the time spent on this review. Please see below how each point has been addressed.

  • Line 53 needs to be reviewed.
    • This typo has been corrected, and now reads: "...thiourea has been commercially incorporated..."
  • In this manuscript, the atomic concentration of elements (Figure 3) extracted from the XPS data is presented with two digits after the decimal. The authors need to show their XPS results are repeatable. Otherwise, only one digit after the decimal is recommended.
    • As repeated XPS analysis is not possible, Figure 3 has now been replaced with an updated figure, which provides atomic concentrations at a more suitable degree of accuracy (1 decimal place).
  • The spectroscopic notation should have the orbitals in italics
    • This has now been corrected throughout Section 3.1.

Reviewer 2 Report

1.This article studied the selectivity and capacity extraction of Cu from mixed-metal acidic solutions by the thiourea-functionalized resin Puromet MTS9140. And The mechanism of extraction was determined to be through reduction of Cu(II) to Cu(I) rather than chelation in this paper. The thiourea-functionalized resin has potential application value in the recovery of copper from mixed-metal acidic solutions. The innovation of this article is finding that the thiourea-functionalized resin Puromet MTS9140 had high selectivity for copper, and the extraction mechanism of Cu by the resin was proposed.

2.This paper could be accepted after minor revision which listed as following,

a, Some results are not marked with error bars such as Figure 13;

b, The subscript of “NaClO3” on line 329 is incorrect;

c, The mechanism of the decrease of thiorea functionalized resin loading capacity of Cu after elution by NaClO3 needs to be further verified. Find out whether the degradation of thiourea resin or CuO passive film hinders the extraction of copper, or both play an important role.

Author Response

The authors thank the reviewer for their constructive comments, and for the time spent completing this review. Our responses to reviewer comments are provided below.

  • Some results are not marked with error bars such as Figure 13
    • Figure 13 has now been replaced with an updated figure which includes the appropriate error bars for IC data. While Figure 12 appears not to have error bars, these are in fact present, but difficult to discern given their small size when compared to the y-axis range.
  • The subscript of “NaClO3” on line 329 is incorrect
    • This typo has now been corrected.
  • The mechanism of the decrease of thiorea functionalized resin loading capacity of Cu after elution by NaClO3 needs to be further verified. Find out whether the degradation of thiourea resin or CuO passive film hinders the extraction of copper, or both play an important role.
    • While the presence of a CuO film could in theory inhibit the adsorption of Cu on the next loading cycle, the first elution recovery percentage of 85.34% suggests that the majority of Cu is removed from the resin surfaces during elution. This, coupled with the observed generation of large amounts of sulfate ions (which occurs prior to Cu elution) instead suggests that it is the degradation of thiourea which is responsible for the reduced capacity in subsequent loading cycles. However, we agree that further validation work is required to confirm these theories, and this explanation has now been added to Section 3.5.

Reviewer 3 Report

The authors have shown here application of thiourea-functionalized ion exchange resin Puromet MTS9140 in selective Copper recovery from a mixed metal stock solution comprising sulfate salts of Al(III), Co(II), 62 Cu(II), Fe(III), Mn(II), Ni(II), and Zn(II).  At line 45-46, it follows that a complex waste stream containing multiple metals would be produced. It should be referenced.

However, Use of thiourea in leaching of Copper group metals (Cu, Ag , Au etc ) is quite well reported in literature. Introduction should be properly supported by previously reported work.

  • There is some mismatch in understanding title of the paper and looking at section 2.1 of Materials and methods.

At line 63 - All metal salts used were of analytical grade and purchased from Sigma-Aldrich. So, where authors have used the multi-component sulfate waste streams, The title is misleading giving an impression that copper is selectively recovered from multi-component sulphate waste streams. Please describe what waste was used, how multi-component sulphate waste streams were generated and how much copper was there in the waste streams, and % extraction of copper from waste stream sample.

The reader would like to know details about waste if waste is referred in the title, how it was leached and % composition of leached metals in leachate including copper which is target metal for recovery.

At line 64-65

“Salts were dissolved in deionized water and acidified to  pH 1 using H2SO4, such that the final concentration of each metal was 2000 mg/L.”  its very important to describe that in real life waste metal composition remains very heterogeneous and complex, one metal can be high and other could be low. There should have been testing of simulated sample corresponding to waste for selective copper extraction.

2) Following point should be also discussed-

Effect of Target metal concentration; and at what concentration of target metal ions of thiourea-functionalized ion exchange resin Puromet MTS9140 gets saturated. Is that saturation concentration dependent of target metal ions -

Author Response

The authors thank the reviewer for their constructive comments, and for the time spent working on this review. Our responses to comments are provided below.

  • Use of thiourea in leaching of Copper group metals (Cu, Ag , Au etc ) is quite well reported in literature. Introduction should be properly supported by previously reported work.
    • Thank you for your comments. The use of thiourea as lixiviant during leaching of copper group metals (Au, Ag) is mentioned in the introduction (from line 49, references 13-15). Given the focus of this work is not on leaching, but on the recovery of metals (specifically Cu) from leachates, it was felt that the attention given to the use of thiourea during the leaching of metals in the introduction was proportionate to the research aims. Literature focused on the use of thiourea-functionalised ion exchange resins for metal extraction, especially Cu, is incredibly limited in its availability.

  • There is some mismatch in understanding title of the paper and looking at section 2.1 of Materials and methods. At line 63 - All metal salts used were of analytical grade and purchased from Sigma-Aldrich. So, where authors have used the multi-component sulfate waste streams, The title is misleading giving an impression that copper is selectively recovered from multi-component sulphate waste streams. Please describe what waste was used, how multi-component sulphate waste streams were generated and how much copper was there in the waste streams, and % extraction of copper from waste stream sample.
    • This work presents the results metal extraction by resins from a synthetic waste stream containing multiple metals. It is apparent that this was not explained well enough, and that the title could be misinterpreted and so this has now been updated in light of your comments.
    • The decision to use a synthetic waste solution (prepared by dissolution of analytical grade metal salts) was to retain exact control over parameters throughout experimentation, and to eliminate the effect that heterogeneous metal concentrations would have on the processes of metal extraction. Rather than focussing on a specific waste leachate, a 'general' leachate composition was used, as determining resin behaviour was the aim of the work rather than treating a specific waste. This aim has now been clarified. Information of solution composition (200 mg/L of all metals), how these were produced (dissolution of metal salts), and the Cu recoveries are presented in the Methods and Results sections, respectively.

  • The reader would like to know details about waste if waste is referred in the title, how it was leached and % composition of leached metals in leachate including copper which is target metal for recovery.
    • As previously addressed, the decision was made not to use a specific leachate from a real waste sample, as the focus of this work (and the wider project) was on resin behaviour rather than engineering a specific waste treatment process for Cu recovery. The title and introductory matter have been updated to clarify that the solutions used in experiments were synthetic rather than being generated from real waste samples.

  • At line 64-65 “Salts were dissolved in deionized water and acidified to  pH 1 using H2SO4, such that the final concentration of each metal was 2000 mg/L.”  its very important to describe that in real life waste metal composition remains very heterogeneous and complex, one metal can be high and other could be low. There should have been testing of simulated sample corresponding to waste for selective copper extraction.
    • As mentioned, the focus of this work was not resource recovery from a specific waste stream, rather the identification of a resin which could be applied to sulfate leachates containing multiple metals. The authors agree that the effect of heterogeneous metal concentrations could be better explained, as this is the rationale for using homogeneous synthetic solutions, and so this has been amended in Section 2.1. 
  • Following point should be also discussed- Effect of Target metal concentration; and at what concentration of target metal ions of thiourea-functionalized ion exchange resin Puromet MTS9140 gets saturated. Is that saturation concentration dependent of target metal ions -
    • The saturation capacity values (or operating capacity (Qo) in this case given the fixed-bed dynamic operation) are reported for studied metals in Tables 2-4. In each instance, minimal extraction of other metals was observed, with only Cu being effectively extracted. However, the point of target metal (Cu) concentration is an important one, and is similar to your previous points regarding heterogeneous metal concentrations in real wastes. While the decision was made here to use one fixed metal concentration, the Conclusion has now been updated to suggest areas of further research to be conducted, which includes the study of real wastes with differing metal concentrations.